# An Eco-Friendly Approach to the Control of Pathogenic Microbes and *Anopheles stephensi* Malarial Vector Using Magnesium Oxide Nanoparticles (Mg-NPs) Fabricated by *Penicillium chrysogenum*

**DOI:** 10.3390/ijms22105096

**Published:** 2021-05-12

**Authors:** Amr Fouda, Mohamed A. Awad, Ahmed M. Eid, Ebrahim Saied, Mohammed G. Barghoth, Mohammed F. Hamza, Mohamed F. Awad, Salah Abdelbary, Saad El-Din Hassan

**Affiliations:** 1Department of Botany and Microbiology, Faculty of Science, Al-Azhar University, Nasr City, Cairo 11884, Egypt; aeidmicrobiology@azhar.edu.eg (A.M.E.); hema_almassry2000@azhar.edu.eg (E.S.); mohamed_gamal.sci@azhar.edu.eg (M.G.B.); abdelbary_salah@azhar.edu.eg (S.A.); 2Department of Zoology and Entomology, Faculty of Science, Al-Azhar University, Nasr City, Cairo 11884, Egypt; Mohamed_awad@azhar.edu.eg; 3Guangxi Key Laboratory of Processing for Non-Ferrous Metals and Featured Materials, School of Resources, Environment and Materials, Guangxi University, Nanning 530004, China; m_fouda21@hotmail.com; 4Nuclear Materials Authority, El-Maadi, Cairo POB 530, Egypt; 5Department of Biology, College of Science, Taif University, P.O. Box 11099, Taif 21944, Saudi Arabia; m.fadl@tu.edu.sa

**Keywords:** green synthesis, MgO-NPs, pathogenic microbes, *Anopheles stephensi*, biocontrol, mosquitocidal

## Abstract

The discovery of eco-friendly, rapid, and cost-effective compounds to control diseases caused by microbes and insects are the main challenges. Herein, the magnesium oxide nanoparticles (MgO-NPs) are successfully fabricated by harnessing the metabolites secreted by *Penicillium chrysogenum*. The fabricated MgO-NPs were characterized using UV-Vis, XRD, TEM, DLS, EDX, FT-IR, and XPS analyses. Data showed the successful formation of crystallographic, spherical, well-dispersed MgO-NPs with sizes of 7–40 nm at a maximum wavelength of 250 nm. The EDX analysis confirms the presence of Mg and O ions as the main components with weight percentages of 13.62% and 7.76%, respectively. The activity of MgO-NPs as an antimicrobial agent was investigated against pathogens *Staphylococcus aureus*, *Bacillus subtilis*, *Pseudomonas aeruginosa*, *Escherichia coli*, and *Candida albicans*, and exhibited zone of inhibitions of 12.0 ± 0.0, 12.7 ± 0.9, 23.3 ± 0.8, 17.7 ± 1.6, and 14.7 ± 0.6 mm respectively, at 200 µg mL^−1^. The activity is decreased by decreasing the MgO-NPs concentration. The biogenic MgO-NPs exhibit high efficacy against different larvae instar and pupa of *Anopheles stephensi*, with LC50 values of 12.5–15.5 ppm for I–IV larvae instar and 16.5 ppm for the pupa. Additionally, 5 mg/cm^2^ of MgO-NPs showed the highest protection percentages against adults of *Anopheles stephensi*, with values of 100% for 150 min and 67.6% ± 1.4% for 210 min.

## 1. Introduction

Nanotechnology is a multidisciplinary science, highly developed during the last two decades to improve and solve a lot of challenges and problems in various fields, such as medicine, environment, agriculture, food, and different industries [1,2]. The new nano-sized materials (1–100 nm) are synthesized based on three basic approaches, including chemical, physical, and biological processes [3]. The synthesized process can be accomplished by two techniques, bottom-up and top-down. The aggregation of atoms to atoms to form new nanomaterials is defined as self-aggregation or bottom-up, whereas the breakdown of starting bulk materials to small or nano-sized is known as the top-down technique [4,5].

The chemical and physical fabrication processes frequently require distinct processing materials and harsh conditions, including unsafe chemicals, pressure, optimized pH, controlled temperature, and expansive instruments. Finally, these fabrication processes are high cost and generate undesirable by-products that cause several diverse ecosystems’ problems [4]. On the contrary, the biological route is characterized by being a simple, rapid, eco-friendly, and low-cost method, as compared to chemical and physical processes [6]. Therefore, researchers’ attention is directed to the green method or biological method using bacteria, fungi, actinomycetes, algae, yeast, and plants to fabricate nanomaterials [7,8,9,10,11]. Recently, there is a huge number of nanomaterials being fabricated using biological methods, such as Ag, Au, Cu, CuO, ZnO, Fe_2_O_3_, TiO_2_, and others [12,13,14,15,16,17]. 

Fungi are considered one of the most biological entities used for the biological synthesis of NPs due to their ability to secrete varied metabolites such as proteins and enzymes which are used for reducing, capping, and stabilizing nanomaterials [18]. Moreover, due to scalability, high efficacy to metal tolerance, easy handling, high biomass production, and economic livability, all these properties give fungi priority to be used as a template for fabricating various nanoparticles [5]. *Penicillium chrysogenum* is the most common fungal species that produces a wide range of metabolites, for instance, fungisporin, roquefortines, penitric acid, siderophores, indole-3-acetic acid, varied enzymes, chrysogenin, ω-hydroxyemodin, and chrysogine [19,20,21]. Therefore, it can be used as a platform to produce different metal and metal oxide nanoparticles. 

The metal oxide nanoparticles have extra advantages such as high chemical stability and less toxicity as compared with other metal nanoparticles [22]. These extra advantages give metal oxides the priority to incorporate into various biomedical and biotechnological applications [23]. Among metal oxide nanoparticles, magnesium oxide nanoparticles (Mg-NPs) have a unique set of desired applications because they possess unique properties, such as photocatalytic characteristics, higher ionic, less toxic to human and animal tissues, and are more stabilized in the body fluids [24]. The MgO-NPs can be integrated into various applications, such as biocatalysts, superconductors, paints, optical imaging, antiadhesion, and wastewater treatment [25,26]. Moreover, they can be used for numerous biological purposes, such as antibiofilm, antifungal, antibacterial, antiviral, antitumor, anticancer, biosensors, heartburn treatments, molecular signaling, bone regeneration, to lower the complications of magnesium shortage, and as additives to killing food-borne pathogens [27,28,29]. MgO-NPs are characterized by low volatility and tolerate high-temperatures, so they can be used as an antimicrobial agent with long-term effects, and interestingly, are proven as a safe nano-compound by the FDA [30,31].

The current study focuses on investigating the multifunctional properties of MgO-NPs fabricated through harnessing metabolites of *Penicillium chrysogenum.* The physicochemical characterizations of biosynthesized MgO-NPs were accomplished using UV-Vis spectroscopy, XRD, TEM, DLS, SEM-EDX, FTIR, and XPS analyses. Moreover, the biological activities of biosynthesized MgO-NPs summarized on antimicrobial activity against pathogenic Gram-positive and Gram-negative bacteria, and unicellular fungi, were investigated. Also, the larvicidal, pupicidal, and repellent activity toward *Anopheles stephensi* malarial vector were assessed.

## 2. Results and Discussion

### 2.1. Isolation and Identification of the Fungal Isolate

The current study adopts an eco-friendly, cost-effective, and biocompatibility approach to the green synthesis of one important metal oxide, magnesium oxide nanoparticles (MgO-NPs). The fungal isolate A2 was isolated from the soil sample and selected for green synthesis of MgO-NPs based on their rapid and greater synthesis. This isolate underwent cultural, microscopic, and molecular identification based on sequence and amplification of the ITS gene. The primary identification based on cultural characteristics showed that the diameter of colonies on malt extract agar media was ranging between 25 and 40 mm, moderate to heavy conidial production, the observed color is dull green, and the reverse color is pale yellow or yellowish-brown (Figure 1A,B). On the other hand, the microscopic examination showed that the conidiophores are usually borne from the surface or subsurface hyphae, and stipes are smooth thin walls of approximately 200 to 300 µm long. The penicilli are 1–2 rami, terverticillate, and usually terminal or subterminal. Phialides appeared with a long diameter of 7–8 µm, while the conidia appeared as a smooth wall of 2.5–4.0 µm long and borne in highly irregular columns (Figure 1C). Based on cultural and microscopic examination, isolate A2 belongs to *Penicillium* spp. [20,32].

Moreover, the sequence analysis of fungal isolate A2 was highly similar to *Penicillium chrysogenum* (accession number: NR077145) with percentages of 97.1%. Hence, the fungal isolate A2 used in the current study was identified as *Penicillium chrysogenum* strain A2 (Figure 1D). The sequence obtained in this study was deposited in Gene Bank under accession number MW774585.

Recently, *Penicillium chrysogenum* strains have been used as a biocatalyst for the green synthesis of a wide range of metal and metal oxide NPs [33,34]. These activities could be attributed to the efficacy of *P. chrysogenum* to produce novel active metabolites which are utilized for reducing, capping, and stabilizing metal and metal oxide NPs [35,36]. To date, this is the first report for the green synthesis of MgO-NPs fabricated by *P. chrysogenum.*

### 2.2. Green Synthesis of MgO-NPs

Fungi are considered a promising tool for the green synthesis of metal and metal oxide NPs due to their reservoir for bioactive compounds. Fungi also tolerate and accumulate a high concentration of metals, and they are characterized by easy handling, biocompatibility, and scalability [5,35]. In the current study, cell filtrate of *P. chrysogenum* A2 participated in the green synthesis of MgO-NPs through reducing Mg^2+^ present in the precursor (Mg (NO_3_)_2_.6H_2_O) by electrons liberated during the reduction of NO_3_ to NO_2_ [37].

At first, the fungal cell-free extract reduced the precursor to form a turbid white precipitate of Mg(OH)_2_, which was collected and rinsed with distilled water to remove any impurities.
(1)$$\mathrm{Mg}{\left({\mathrm{NO}}_{3}\right)}_{2}.6H_{2}O+H_{2}O\overset{\mathrm{Fungal}}{\underset{\mathrm{Metabolites}}{\to }}\mathrm{Mg}{\left(OH\right)}_{2}$$

After that, the as-formed Mg(OH)_2_ was calcinated at 400 °C for 4 h to form MgO-NPs, as reported by Essien et al. [38]:(2)$$\mathrm{Mg}{\left(OH\right)}_{2}\overset{400{\;}^{o}C}{\to }\;\mathrm{MgO}-\mathrm{NPs}$$

### 2.3. Characterizations of Biosynthesized MgO-NPs

#### 2.3.1. UV-Vis Spectroscopy Analysis

The color intensity as an indicator for MgO-NPs synthesis was monitored by detecting the maximum surface plasmon resonance (SPR) using UV-Vis spectroscopy. Data analysis showed that the maximum SPR for MgO-NPs synthesized by *P. chrysogenum* A2 was 250 nm (Figure 2A). The morphological characteristics of NPs such as size, shape, and distributions are usually correlated with their SPR, as reported previously [39]. Moreover, the size of MgO-NPs was small or large according to the SPR values of less or more than 300 nm, respectively [40]. Interestingly, the SPR of MgO-NPs synthesized by *Rosmarinus officinalis* L. was observed at 250 nm [41]. Also, Essien et al. [38] showed that the SPR of MgO-NPs fabricated by *Manihot esculenta* was 260 nm. Based on obtained data, it can be concluded that metabolites secreted by *P. chrysogenum* A2 were efficient to fabricate MgO at the nanoscale.

#### 2.3.2. X-Ray Diffraction (XRD)

The XRD analysis is used to detect or measure the crystalline nature of green synthesized MgO-NPs. The XRD pattern showed five sharp peaks at 2θ° of 36.8°, 42.7°, 62.1°, 46.4°, and 78.5°, which indexed to (111), (200), (220), (311), and (222) planes (Figure 2B). The presence of some little peaks indicates that the sample has some impurities [41]. The observed peaks confirm the successful formation of the crystallographic phase of face-centered-cubic (FCC) structure according to JCPDS file No. (89-7746) [42]. The XRD pattern confirms the presence of Mg(OH)_2_ and MgO in the sample, and this finding is confirmed by XPS analysis. The observed plan peaks of (111) and (311) at 2θ° values of 36.8° and 46.4° refer to Mg(OH)_2_, whereas plan peaks of (200), (220), and (222) refer to MgO [43]. In most cases, the final product contains Mg(OH)_2_ and MgO, and can eliminate the Mg(OH)_2_ by elevating the temperature to 700 °C as previously reported [44]. The crystal size of NPs can be calculated from the XRD pattern using the Debye–Scherrer equation. Data showed that the average crystal size of MgO-NPs was 48 nm. Recently, the particle size of MgO-NPs synthesized by harnessing metabolites of *Aspergillus carbonarious* D-1 calculated using the Debye–Scherrer equation was 35 nm [45], whereas those synthesized by leaf extract of *Azadirachta indica* was 27 nm [46].

#### 2.3.3. Transmission Electron Microscopy (TEM)

TEM analysis is a useful technique to investigate the morphological characteristics of NPs, such as shape, size, and agglomeration percentages [47]. In this study, the TEM image showed the efficacy of metabolites secreted by *P. chrysogenum* to fabricate spherical MgO with a size range of 7–40 nm, with an average diameter of 16.9 ± 7.5 nm (Figure 2C,D). The TEM image exhibits the well-dispersed green synthesized MgO-NPs without any aggregation. The obtained data are consistent with Bindhu et al. [48], who successfully synthesized well-dispersed, spherical MgO-NPs, with a size range of 7–38 nm and an average size of 16 nm. The biological activities of NPs are known to correlate with their size, and if the size is decreased, the activities are increased [5]. For example, MgO-NPs with varied sizes of 35.9, 47.3, and 2145.9 nm showed different potential to inhibit the growth of *Bacillus subtilis*, with percentages of 96.1%, 94.5%, and 75.7%, respectively [49]. Therefore, based on the obtained size of green synthesized MgO-NPs in the current study, we predict that the activities are high.

#### 2.3.4. Dynamic Light Scattering (DLS) Analysis

The nanoparticles’ size, as well as the size distribution of biosynthesized MgO-NPs, were analyzed using the DLS technique based on hydrodynamic diameters. The histogram of the particle size distribution (Figure 3A) showed that the average hydrodynamic particle diameters were 55.8 and 20.13 nm for volume intensity of 2.3% and 97.7%. The average size obtained by the DLS technique is bigger than those recorded by TEM and XRD because of coating metabolites on the NPs surface, which are used for capping and stabilizing MgO-NPs [50,51]. Additionally, the high diameter size of NPs measured by DLS could be attributed to the non-homogeneous distribution of particles in the colloidal solution [52,53].

The DLS analysis provides more information about the homogeneity of particles into the colloidal solution by measuring the polydispersity value (PDI) [4]. The homogeneity was increased or decreased according to the PDI value, where it increased if the PDI value is less than 0.4 and decreased if the PDI value is more than 0.4, as reported previously [54]. If the PDI value is more than 1.0, the solution becomes highly heterogeneous. In the current study, the PDI value of biosynthesized MgO-NPs was 0.3, which indicates the homogeneity of MgO-NPs colloidal solution.

#### 2.3.5. Energy Dispersive X-ray (EDX) Analysis

The qualitative and quantitative compositions of as-formed MgO-NPs were analyzed using EDX analysis. The EDX graph demonstrated the presence of Mg and O elements in the sample, which confirms the successful fabrication of MgO by metabolites secreted by *P. chrysogenum.* The presence of Mg and O peaks at bending energy between 0.5 and 1.5 KeV indicate the successful formation of MgO [55]. The quantitative analysis confirms the presence of Mg and O with weight percentages of 13.62% and 7.76% respectively, while the atomic percentages of Mg and O ions in the sample were 11.18% and 5.12%, respectively (Figure 3B). The presence of other peaks in EDX spectra indicates the presence of some impurities in the sample, which is confirmed by XRD and XPS analyses. The weight percentages of C, Cl, and Ca ions were 75.33%, 1.17%, and 2.12%, respectively. Consistent with our study, Dobrucka [55] reported the presence of additional peaks, including Al, Si, K, and Ca, besides Mg and O during phytosynthesis of MgO-NPs by *Artemisia abrotanum.* These additional peaks could be attributed to the breakdown of metabolites around the biosynthesized MgO-NPs by X-ray emission [56,57].

#### 2.3.6. Fourier Transform Infrared (FT-IR) Spectroscopy

The roles of functional groups present in fungal biomass filtrate in reducing and forming of MgO-NPs were analyzed using FT-IR, which was measured at wavenumbers of 400–4000 cm^−1^ (Figure 3C). The sharp peak at 3695 cm^−1^ was assigned to the OH stretching band [58], while the broadband observed at 3420 cm^−1^ was assigned to the O-H stretching of hydroxyls overlapped with N-H stretching vibration of amines found in the polysaccharide [55,59]. A series of low resolved peaks that appeared in the range 2900–2700 cm^−1^ (overlapped with the broadness peak of N-H and O-H stretching band) were assigned to C-H stretching of aliphatic hydrocarbons with methoxy CH_3_-O- (i.e., 2845 cm^−1^) [60], and also, the band that appeared at 2720 cm^−1^ was mainly assigned to methylamine N-CH_3_ and C-H stretching [60]. These results were identical with the data obtained from the XPS analysis for C 1s, O 1s, and N 1s binding energies.

A high resolved peak with broadness that appeared at 1635 cm^−1^ was assigned to C=O of amide, carboxylate (carboxylic acid salt), and N-H stretching vibration, while the other peak at 1415 cm^−1^ related to bending of C-H bonds, and C=O of carboxylate [61]. The peak at 1165 cm^−1^ was assigned to the C-O stretch of the alcohol that was present in the hydrocarbon skeletons, which overlapped with C-N stretching of tertiary amine (see the XPS analysis) [60]. The low resolved peak at 1030 cm^−1^ was related to C-H out-of-plane bend, C-O stretching, CN stretch of primary amine [60,62], and Mg–OH stretching [63]. Peaks were found at 650 and 520 cm^−1^ in the figure print region, identical to the β-D-glucose unit (from the hydrocarbons), bending of free amine bond [62,64], as well as Mg-O vibration peaks, which verify the successful fabrication of these particles [58,65]. Based on FT-IR analysis, the presence of different functional groups for polysaccharides, hydrocarbons, amines, carboxylate, and amino groups confirms the efficacy of metabolites involved in fungal biomass filtrate for reducing, capping, and stabilizing MgO-NPs.

#### 2.3.7. X-ray Photoelectron Spectroscopy (XPS) Analysis

The characterization of the MgO NP was completed by the analysis of XPS. Figure 4A shows the overall survey spectra which studied the selected binding energy ranges. This survey showed binding energies of C 1s, O (1s 2s, KL1, and KL2), N 1s, and Cl (1s, 2p), while different species of Mg were detected (i.e., 2s, 2p, KL1, KL2, KL3, KL4, KL5, and 1s), indicating the majority of this component over others.

The chemical compositions are clearly shown when comparing the different profiles of the signals as well as their deconvolutions. The C 1s spectra (Figure 4B) shows five splitting peaks for C(C, N, H), C(=N, O), or C-O-C, N-C=O (amide), O-C=O, and O-C-O at 284.07, 285.48, 287.31, and 289.89 eV respectively [62,66,67], and this gives good evidence for the carbohydrate produced by the organisms. The O 1s spectra (Figure 4C) shows two internal splitting peaks at 530.82 and 532.27 eV, which are assigned for O(N, C, H), O-C=O [68,69], and this supports the data produced by the C 1s deconvolutions. The N 1s (Figure 4D) was deconvoluted into five internal peaks at 398.87 eV for N (C, H) and 400.34 eV for N_tert_ of polysaccharides [70], while the medium reflects three peaks for NO, NO_2_, and NO_3_ at 402.8, 405.77, and 406.68 eV respectively, resulted from the source of nitrate used for dissolving the salt [71].

The XPS spectra of Mg shows predominately MgO over Mg(OH)_2_, and these were shown by the deconvolution of Mg 1s, which produced two peaks at 1304.44 eV (major) for MgO with At. 94.74%, and the other peak (5.26 At%; minor) at 1306.28 eV [72]. The Mg 2p also shows two peaks for MgO and Mg-OH at 49.49 and 48.74 eV, with At. 95.89% and 4.11%, respectively [73,74]. Again, progressively more increased MgO than Mg (OH)_2_ was shown in the Mg 2s spectra, which deconvoluted into two peaks at 88.24 eV (At. 91.77%) and 87.61 eV (At. 8.23%) for MgO and Mg(OH)_2_, respectively [74]. From this data, it was clear that MgO is the main species in composite materials.

### 2.4. Antimicrobial Activity

The use of various antibiotics has led to the spread of microbial resistance; therefore, it is necessary to discover new compounds that have antimicrobial activity. Biosynthesized MgO-NPs have unique properties, making them a good source to develop new antimicrobial compounds [75]. In the current study, the potentiality of biosynthesized MgO-NPs to inhibit the pathogenic Gram-positive and Gram-negative bacteria, and unicellular fungi, was investigated using the agar well diffusion method. Data analysis showed that the activities of biosynthesized MgO-NPs as antibacterial and anti-*candida* were dose-dependent, where the activities increased by increasing the NPs concentrations. These findings were compatible with published studies about nanomaterials as antimicrobial agents [15,76,77]. The zones of inhibition (ZOI) formed due to the highest MgO-NPs concentration (200 µg mL^−1^) were 12.7 ± 0.9, 12.0 ± 0.0, 23.3 ± 0.8, 17.7 ± 1.6, and 14.7 ± 0.6 mm for *Bacillus subtilis*, *Staphylococcus aureus*, *Pseudomonas aeruginosa*, *Escherichia coli*, and *Candida albicans*, respectively (Figure 5). Umaralikhan and Jaffar [78] reported that the ZOI formed due to treatment with 5 mg mL^−1^ of MgO-NPs synthesized by *Pisidium guvajava* was 16 and 15 mm for *S. aureus* and *E. coli* respectively, while those synthesized by *Aloe vera* formed ZOI with the value of 15 and 12 mm against the same organisms.

The MIC values, which are defined as the minimum MgO concentration that can inhibit the microbial growth, were assessed through examination of various concentrations (100, 50, 25 µg mL^−1^). Data showed that the MIC value for *B. subtilis*, *S. aureus*, and *C. albicans* was 100 µg mL^−1^, with ZOIs of 8.7 ± 0.6, 8.2 ± 0.3, and 9.3 ± 0.9 mm, respectively. Whereas the MIC value for Gram-negative bacteria, *P. aeruginosa*, and *E. coli* was 50 µg mL^−1^, with ZOIs of 9.7 ± 0.5 and 8.2 ± 0.3 mm, respectively.

Based on obtained data, the Gram-negative bacteria are more sensitive to biosynthesized MgO-NPs than Gram-positive bacteria. This activity can be attributed to the difference in the cell wall structure between the two kinds. The cell wall of Gram-positive bacteria contains a thick layer of peptidoglycans, in contrast to Gram-negative bacteria, which contain a thin layer of peptidoglycans and extra lipopolysaccharides (LIP). The attraction between NPs and bacterial cell wall was due to the negative charge on the LIP and positive charge on the NPs surface [79]. Additionally, because of the small size of biosynthesized MgO-NPs, which in the current study was 16.9 ± 7.5 nm, they can penetrate the thin peptidoglycan layer easily and hence disrupt the selective permeability functions of the microbial cell membrane [80]. Moreover, the entrance of MgO-NPs into the microbial cell led to blocking the quorum sensing between cells and hence, inhibited the cellular functions [81].

Besides the previous mechanism, some authors regard the antimicrobial activity of MgO-NPs to the production of reactive oxygen species (ROS), dissociation of Mg^2+^ ions inside the microbial cells, and alkaline effects. The ROS enhances the formation of hydrogen peroxide (H_2_O_2_), superoxide radicals (^−^O_2_), and reactive hydroxyl radical (^•^OH), which react with nucleic acids and proteins and hence destroy the internal cellular components [63]. The dissociated Mg^2+^ ion inside the cells reacts with the -SH group of amino acids and leads to the breakdown of protein structure, and ultimately to cell death [65]. Finally, the water vapor condenses on the MgO-NPs surface, forming a thin water zone characterized by a greater pH value than the equilibrium state. Once MgO-NPs react with microbial cells, the greater pH in the formed water zone leads to destroying the cell wall and cell membranes, ending in cell death [82].

### 2.5. Larvicidal/Pupicidal Bioassay

Mosquitoes are considered the main carriers of pathogens in most equatorial and subtropical countries. Mosquitoes are the causative agent for malaria, dengue fever, yellow fever, filariasis, chikungunya, encephalitis, and others [83]. The most common insecticides used are synthetic compounds that disrupt a biological system due to repeated and uncontrollable usage. Therefore, it is urgent to discover new eco-friendly active compounds that have mosquitocidal properties. Herein, the efficacy of biosynthesized MgO-NPs as larvicidal for *Anopheles stephensi* was investigated. Data analysis showed that the potentiality of MgO-NPs as larvicidal and pupicidal was dose- and time-dependent. Data represented in Table 1 show that the percentages of larvae mortality increased from 30.2% ± 1.09% (I instar), 29.6% ± 1.14% (II instar), 27.8% ± 1.30% (III instar), and 24.4% ± 1.94% (IV instar) at 5 ppm to 91.8% ± 2.38% (I instar), 88.2% ± 1.64% (II instar), 81.4% ± 0.89% (III instar), and 72.8% ± 2.58% (IV instar) at 25 ppm. Moreover, the LC50 (the concentration of MgO-NPs that causes 50% mortality) increased from 12.4 ppm for I instar to 15.6 ppm for IV instar, while LC90 (the concentration of MgO-NPs that causes 90% mortality) increased from 22.3 ppm for I instar to reach 27.9 ppm for IV instar. On the other hand, the mortality percentages of pupa due to treatment with the highest MgO-NPs concentration (25 ppm) was 69.2% ± 2.8%, with LC50 of 16.5 ppm and LC90 of 29.8 ppm.

The efficacy of nanoparticles against different mosquitos was previously investigated. Madhiyazhagan et al. [84] reported the 30 ppm of Ag-NPs synthesized by extract of *Sargassum muticum*, causing mortality for first to fourth instar and pupa, with percentages of 69.6% ± 2.5%, 66.2% ± 1.9%, 61.8% ± 2.9%, 53.4% ± 1.5%, and 50.2% ± 1.5% for *Aedes aegypti*, and 59.6% ± 2.3%, 57.8% ± 1.8%, 55.4% ± 1.8%, 51.2% ± 2.3%, and 48.4% ± 0.9% for *Culex quinquefasciatus.* Moreover, the LC50 values of TiO_2_-NPs synthesized by *Argemone mexicana* against different instar and pupa of *Aedes aegypti* were 17.9 ppm (I instar), 21.7 ppm (II), 26.1 ppm (III), 30.04 ppm (IV), and 35.3 ppm (pupa), while the LC90 values were 41.6 ppm (I instar), 51.1 ppm (II), 56.5 ppm (III), 62.9 ppm (IV), and 71.7 ppm (pupa) [85]. The LC50 values of bio-encapsulated chitosan/silver nano-complex were ranging between 54.65 and 98.17 ppm, which exhibits low larvicidal and pupicidal properties as compared with non-encapsulated chitosan/silver (LC50 ranging between 4.4 and 7.6 ppm) [86]. By comparing the obtained data with published studies, it can be concluded that the biosynthesized MgO-NPs were more toxic against different instar larvae and pupa at low concentrations. To the best of our knowledge, this is the first report to study the efficacy of fungal-mediated biosynthesis of MgO-NPs against *Anopheles stephensi* malaria vector.

The mechanisms of MgO-NPs as mosquitocidal can be summarized into two points, first is their efficacy to produce reactive oxygen species (ROS) and second is their efficacy to damage the cell wall [87]. Interestingly, MgO-NPs are characterized by their efficacy to produce a high amount of ROS as compared to other metal oxide NPs, while being low in toxicity to humans, plants, and animals [88,89]. Therefore, MgO-NPs can be candidates as mosquitocidal agents to use in the agricultural sector to prevent or reduce the insect population through secretion of ROS without any adverse impacts on the environment and their beings. Upon contact of MgO-NPs with one stage of the *A. stephensi* life cycle, it breaks down into Mg^2+^ and O^2−^ ions in the surrounding environment. The oxidative stress and peroxidation of lipid are formed due to the formation of ROS because of increased O^2−^ ion concentrations (Figure 6) [90]. Moreover, the cellular components are discharged due to destabilization of cellular equilibrium due to an increase in the concentration of Mg^2+^, which ultimately leads to mosquito cell death (Figure 6) [65]. Some researchers reported that the nanoparticles react with the thiol group of amino acids or phosphate group in nucleic acid and hence deformed it, which ultimately inhibits the cell function [26,84].

### 2.6. Repellent Activity

Recently, various synthetic compounds have been analyzed to investigate their efficacy as a repellent agent against mosquitoes. However, the high cost of some proprietary chemical formulations as mosquito repellent agents, such as DEET (N, N-diethyl-m-toluamide), decrease their usage in low-income countries [91]. Therefore, the discovery of eco-friendly and cost-effective repellent compounds is considered the main challenge. In the current study, the efficacy of biosynthesized MgO-NPs as a repellent agent was investigated. Data analysis showed that the positive control (EDDT) exhibited 100% repellent percentages at the end of the experiment. On the other hand, all concentrations of biosynthesized MgO-NPs exhibited 100% repellency until 120 min, and the activities decreased with time (Table 2). At MgO-NPs concentration of 5 mg/com^2^, the repellency reached 100% after 150 min and 67.6% ± 1.4% after 210 min, while at 10 mg/cm^2^, the repellency reached 80.9% ± 1.9% and 59.6% ± 1.5% after 120 and 210 min, respectively. Based on obtained data, it can be recommended that the optimum MgO-NPs concentration used as a repellent agent against *A. stephensi* malaria vector is 5 mg/cm^2^.

Recently, various substances were tested as repellent agents against different mosquitos, for instance, different concentrations (1, 2.5, and 5 mg/cm^2^) of *Citrullus vulgaris* extract obtained by a different solvent system, including methanol, petroleum ether, benzene, and ethyl acetate, exhibited 100% protection for times ranging between 112 to 387 min against *A. stephensi* [92]. Additionally, 100 ppm of TiO_2_-NPs exhibited 80.4% protection against *Aedes aegypti* [85]. The current study provides a new eco-friendly, rapid, and low-cost approach for control of various larvae instar or pupa of *A. stephensi* malarial vector at low concentration, and it can also be used as a repellent agent for complete protection for 150 min at low concentration.

## 3. Materials and Methods

### 3.1. Chemicals Used

All chemicals in this study were analytical grade and procured from Sigma Aldrich, Cairo, Egypt. The magnesium nitrate hexahydrate (Mg(NO_3_)_2_.6H_2_O) was used as a precursor for magnesium oxide nanoparticles (MgO-NPs). The different media types, Malt Extract agar (MEA) media for fungal isolation and cultivation and Muller Hinton agar media for antimicrobial activities, were ready-made (Oxoid^TM^, Thermo-Fisher Scientific, USA). The unicellular fungi were grown in yeast extract peptone dextrose (YEPD) agar media (containing g L^−1^: glucose, 20; peptone, 20; yeast extract, 10; Agar, 20; distilled water, 1000 mL). All biological reactions were achieved using distilled water (dis. H_2_O).

### 3.2. Isolation and Identification of the Fungal Strain

The soil sample used for isolation of the fungal strain was collected from Giza Governorate, Egypt (E: 31°24’62.81”, N: 29°79’32.09”). The isolation procedures were accomplished according to Fouda et al. [93] as follows: 1.0 g of collected soil was diluted in sterilized dis. H_2_O. About 100 µL of the fourth dilution was plated onto MEA plates and incubated for 3–4 days at 30 ± 2 °C. the colonies that appeared were picked up and re-inoculated onto the same media for purifications. The purified colony was preserved on an MEA slant for further use.

The identification was accomplished by cultural and microscopic characters and confirmed using molecular identification using internal transcribed spacer (ITS) sequence analysis. The ITS rDNA region was amplified using primers for ITS1-F (5-CTTGGTCATTTAGAGGAAGTAA-3) and ITS4 (5-TCCTCCGCTTATTGATATGC-3) [94]. The PCR mixture contained: 1X PCR buffer, 0.5 mM MgCl_2_, 2.5 U Taq DNA polymerase (QIAGEN, Germantown, MD 20874, USA), 0.25 mM dNTP, 0.5 µL of each primer, and 1 µg of extracted genomic DNA. The PCR was performed in a DNA Engine Thermal Cycler (PTC-200, BIO-RAD, Hercules, CA, USA) with a program of 94 °C for 3 min, followed by 30 cycles of 94 °C for 30 s, 55 °C for 30 s, and 72 °C for 1 min, followed by a final extension performed at 72 °C for 10 min. The PCR product was checked for the expected sizes on a 1% agarose gel and was sequenced by Sigma Company for scientific research, Egypt, with the two primers. The sequence was compared against the GenBank database using the NCBI BLAST tool. Multiple sequence alignment was performed using the Clustal Omega software package (https://www.ebi.ac.uk/Tools/msa/clustalo, last modified on 1 October 2019 ) and a phylogenetic tree was constructed using the neighbor-joining method with MEGA (Version 6.1) software (MEGA, Auckland, New Zeland), with confidence tested by bootstrap analysis (1000 repeats).

### 3.3. Green Synthesis of MgO-NPs

The purified fungal strains were inoculated into 100 mL of malt extract broth (MAB) media and incubated for 5 days at 30 ± 2 °C and shaking state at 150 rpm. The fungal biomass was collected at the end of the incubation period through centrifugation of inoculated MAB media at 1000 rpm for 5 min. After that, the collected fungal biomass (10.0 g) was resuspended in 100 mL dis. H_2_O for 48 h at 30 ± 2 °C and shaking state at 150 rpm. The previous suspensions were centrifuged at 10,000 rpm for 5 min. The upper layer (fungal biomass filtrate) was collected and used for green synthesis of MgO-NPs as follows: About 76.9 mg of Mg(NO_3_)_2_.6H_2_O was dissolved in 10 mL dis. H_2_O and mixed with 90 mL of fungal biomass filtrate and incubated for 24 h, to get a final concentration of 3 mM. At the end of the incubation period, the turbid white precipitate was collected and rinsed with dis. H_2_O to remove any impurities before being oven-dried at 400 °C for 3 h [26].

### 3.4. Characterization of Biosynthesized MgO-NPs

#### 3.4.1. UV-Vis Spectroscopy

The intensity of color formed as a result of MgO-NPs formation was detected using UV-Vis spectroscopic analysis at a wavelength of 150–500 nm. The surface plasmon resonance (SPR) was detected at the maximum observed peak. The absorbance was measured using a Jenway 6305 Spectrophotometer.

#### 3.4.2. X-ray Diffraction (XRD)

The crystalline structure of fungal-mediated MgO-NPs synthesis was assessed using XRD analysis by X’Pert pro diffractometer (Philips, Eindhoven, Netherlands). The operating conditions were: 2θ values measured in ranges of 4° to 80°, X-ray radiation source was Ni-filtered Cu Ka, and the operating voltage and current were 40 KV and 30 mA, respectively. The average MgO-NPs sizes were measured using the Debye–Scherrer equation [95] as follows:(3)D=Kλ/βCosθ
where, D is average particle size, K is the Scherrer’s’ constant (0.9), λ is the wavelength of X-ray radiation (0.154 nm), and β and θ are the half of maximum intensity and Bragg’s angle, respectively.

#### 3.4.3. Transmission Electron Microscopy (TEM)

The shapes and sizes of biosynthesized MgO-NPs were investigated using TEM analysis (JEOL 1010, Japan, acceleration voltage of 120 KV). A drop of MgO-NPs solution was loaded on the carbon-copper grid and it underwent vacuum desiccation for 24 h, and after that, was placed onto a specimen holder [96].

#### 3.4.4. Dynamic Light Scattering (DLS)

The distribution and sizes of MgO-NPs in colloidal solution were detected by DLS analysis. The sample was subjected to measurement by Zeta sizer nano series (Nano ZS), Malvern, UK.

#### 3.4.5. Energy Dispersive X-ray (EDX) Analysis

The qualitative and quantitative composition of MgO-NPs were assessed using energy dispersive X-ray (SEM-EDX) (JEOL, JSM-6360LA, Tokio, Japan).

#### 3.4.6. Fourier Transform Infrared (FT-IR)

The functional groups present in fungal biomass filtrate and involved in reducing, capping, and stabilizing MgO-NPs were investigated using Fourier transform infrared (FT-IR) spectroscopy (Agilent system Cary 660 FT-IR model, Agilent, Santa Clara, CA, USA). The MgO-NPs sample was mixed with KBr and pressured to form a disk that scanned in the range of 400 to 4000 cm^−1^.

#### 3.4.7. X-Ray Photoelectron Spectroscopy (XPS)

The X-ray photoelectron spectroscopy (XPS) analyses were analyzed by an ESCALAB 250XI^+^ instrument (Thermo Fischer Scientific, Inc., Waltham, MA, USA) connected with monochromatic X-ray Al Kα radiation (1486.6 eV). The analysis was conducted under the following conditions: the size of the spot was 500 µm, the samples were prepared under a pressure adjusted to 10^−8^ mbar, the energy was calibrated with Ag3d_5/2_ signal (∆BE: 0.45 eV) and C 1s signal (∆BE: 0.82 eV), and the full- and narrow-spectrum pass energies were 50 and 20 eV, respectively [66,97].

### 3.5. Antimicrobial Activity

The antimicrobial activity of MgO-NPs synthesized by fungal metabolites was investigated against Gram-positive pathogenic microbes (*Staphylococcus aureus* ATCC 6538, *Bacillus subtilis* ATCC 6633), Gram-negative pathogenic microbes (*Pseudomonas aeruginosa* ATCC 9022, *Escherichia coli* ATCC 8739), and unicellular fungi (*Candida albicans* ATCC 10231) [15]. A 100 mL of Muller Hinton agar media (for bacterial growth) and yeast extract peptone dextrose (YEPD) agar media (for *C. albicans* growth) was inoculated by 50 µL of overnight culture. The inoculated media were poured into sterilized Petri plates under sterilized conditions. Three wells (0.7 cm diameter) were cut in the seeded Muller Hinton or YEPD agar plates and filled with 100 µL of biosynthesized MgO-NPs (200 µg mL^−1^). Different concentrations of MgO-NPs (150, 100, 50, and 25 µg mL^−1^) were prepared to detect the minimum inhibitory concentration (MIC). The loaded Muller Hinton plates were kept in the refrigerator for 1 h before incubation at 35 °C for 24 h. The results were recorded as zones of inhibition (ZOIs) around each well by mm [8]. The experiments were performed in triplicate.

### 3.6. Mosquitocidal Assay

#### 3.6.1. Rearing of Anopheles stephensi

The larvae of *Anopheles stephensi* were purchased from the Medical Entomology Lab., Dokki, Giza, Egypt, and subjected to rearing again in the Medical Entomology Lab., Animal-House, Department of Zoology and Entomology, Faculty of Science, Al-Azhar University, Cairo, Egypt.

Larvae to adult mosquitoes were reared at 27 ± 2 °C, 75–85% relative humidity (RH), and a photoperiod of 14: 10 light: dark condition. The larvae were fed daily on 5 g of 3: 1 ground dog biscuit: brewer’s yeast. The emerging pupae were collected and put into a plastic cup containing 500 mL of water and transferred to inside a screen cage (90 cm length × 90 cm height × 90 cm width) until adults emerged, which fed on 10% (*w*/*v*) sucrose (replaced every two days to avoid the fungal growth). Five-day emergence adults were starved for 12–24 h and then allowed to be fed on the cattle-originated blood. About 5 mL of cattle blood was poured onto the membrane feeder and the blood temperature was adjusted to 37 °C. The blood membrane feeder was placed into the cage for 1–2 h and removed. The egg collection cup containing filter paper and 10 mL distilled water was put inside the cage for oviposition by female mosquitoes [98].

#### 3.6.2. Larvicidal/Pupicidal Assay

The toxicity of different concentrations (25, 20, 15, 10, and 5 µg mL^−1^) of biosynthesized MgO-NPs was investigated against various larvae instar (I, II, III, or IV instars) and pupa of *A. stephensi.* Briefly, 25 larvae or pupa were put into the glass cup containing 500 mL dechlorinated water and supplemented with 5 mL of MgO-NPs (25 µg mL^−1^) and 0.5 mg of the larvae food. The previous step was repeated for each MgO-NPs concentration and each experiment was repeated five times with all larvae instars and pupae. The control was run alongside each experiment without adding MgO-NPs. The mortality percentages (%) were calculated after 24 h according to the following equation:(4)$$\mathrm{Mortality}\;\mathrm{percentages}\;(\%)\;=\;\frac{\mathrm{Number}\;\mathrm{of}\;\mathrm{dead}\;\mathrm{individuals}}{\mathrm{Number}\;\mathrm{of}\;\mathrm{treated}\;\mathrm{individuals}}\times 100$$

#### 3.6.3. Repellent Activity

The repellent activity of different prepared MgO-NPs concentrations (10, 7.5, 5, 2.5, and 1.0 mg cm^−1^) was assessed according to the method of Murugan et al. [85] using a cotton pad. Briefly, the cotton pad was firstly soaked in cattle-originated blood, followed by painting with different prepared MgO-NPs concentrations, separately. Another blood cotton pad was painted with DEET (N, N, diethyl-meta-toulamide) (Johnson Wax Egypt), which served as a positive control. The treated and control cotton pads were placed into the standard cage (30 × 30 × 30 cm^3^) containing 50 *A. stephensi* starved females for 210 min. The experiment was repeated three times for each concentration used. The repellency of treated and control cotton pads was calculated after 15, 30, 60, 90, 120, 150, 180, and 210 min using the following equation:(5)$$\mathrm{Repellency}\;\mathrm{percentages}\;(\%)\;=\;\frac{C-T}{C}\times 100$$
where C denotes the total number of mosquitos before treatment, and T denotes the total number of mosquitos after treatment.

### 3.7. Statistical Analysis

All results presented here are the means of three independent replicates. Data were subjected to statistical analysis by the statistical package SPSS v17. The mean difference comparison between the treatments was analyzed by *t*-test or the analysis of variance (ANOVA) and subsequently, by Tukey’s HSD test, at *p* < 0.05.

## 4. Conclusions

In the current study, the metabolites secreted by *Penicillium chrysogenum* strain A2 were used for reducing, capping, and stabilizing MgO-NPs. The characterization of as-formed MgO-NPs was accomplished using UV-Vis, XRD, TEM, DLS, EDX, FT-IR, and XPS analyses. The maximum SPR for biosynthesized MgO-NPs was observed at 250 nm. Additionally, a crystalline nature, well-dispersed, spherical shape with an average size of 16.9 ± 7.5 nm was detected by XRD, DLS, and TEM analyses. The presence of Mg and O ions at various pending energies was confirmed by EDX and XPS analyses. Data showed that the biological activities of biosynthesized MgO-NPs were dependent on time and concentration. The biogenic MgO-NPs showed antimicrobial activities against pathogenic microbes represented by *Staphylococcus aureus*, *Bacillus subtilis*, *Pseudomonas aeruginosa*, *Escherichia coli*, and *Candida albicans*, with a varied zone of inhibition. Data analysis showed that Gram-negative bacteria were more sensitive to MgO-NPs than Gram-positive bacteria and unicellular fungi. The MIC value for *B. subtilis*, *S. aureus*, and *C. albicans* was 100 µg mL^−1^, with ZOIs of 8.7 ± 0.6, 8.2 ± 0.3, and 9.3 ± 0.9 mm, respectively. Whereas the MIC value for Gram-negative bacteria, *P. aeruginosa*, and *E. coli* was 50 µg mL^−1^, with ZOIs of 9.7 ± 0.5 and 8.2 ± 0.3 mm, respectively. Moreover, the biogenic MgO-NPs exhibited larvicidal, pupicidal, and repellent activity against *Anopheles stephensi* malaria vector at low concentrations. The LC50 and LC90 values for different larva instar (I–IV) ranged between 12.4 and 15.6 ppm and 22.3 and 27.9 ppm, respectively. Additionally, the LC50 and LC90 values for pupa were 16.5 and 29.8 ppm, respectively. Interestingly, the MgO-NPs concentration of 5 mg cm^2^ exhibited the highest repellent percentages, of 100% after 150 min and 67.6% ± 1.4% after 210 min. According to the obtained data, it can be concluded that *P. chrysogenum* has the ability to form MgO-NPs which exhibit high potentiality to control pathogenic microbes and malarial vector insects.

## Figures and Tables

**Figure 1 ijms-22-05096-f001:**
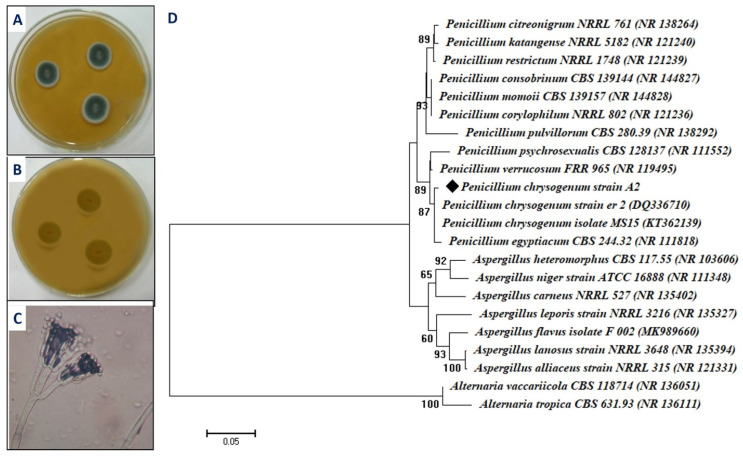
Cultural, microscopic, and ITS sequence analysis of fungal isolate A2. (**A**) Colony of fungal isolate A2 on malt extract agar media, (**B**) reverse colony of isolate A2 on malt extract agar media, (**C**) bright-field microscopic examination (X = 800), and (**D**) phylogenetic tree of isolate A2 with the sequences from NCBI. The symbol ◆ refers to ITS fragments retrieved from this study. The tree was conducted with MEGA 6.1 using the neighbor-joining method.

**Figure 2 ijms-22-05096-f002:**
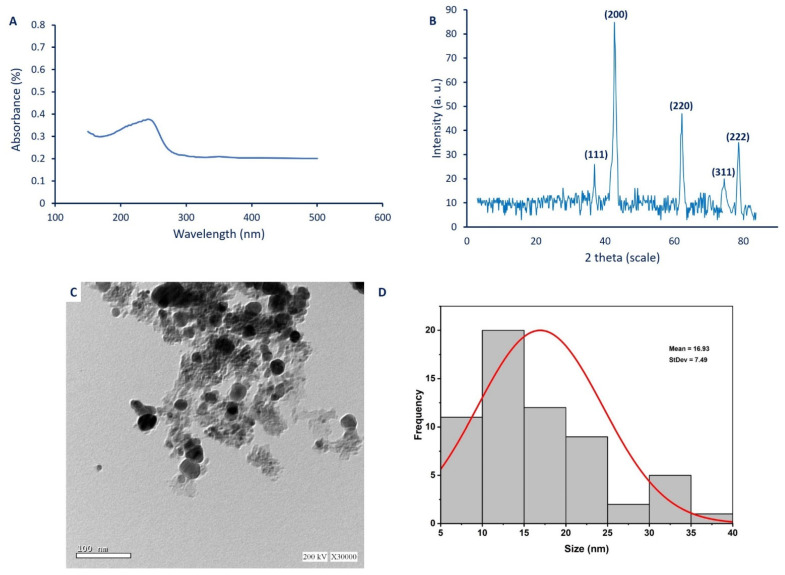
Characterization of biosynthesized MgO-NPs. (**A**) UV-Vis spectroscopy showed maximum SPR at 250 nm, (**B**) XRD analysis showed crystalline nature of biosynthesized MgO-NPs, (**C**) TEM image, and (**D**) size distribution of MgO-NPs.

**Figure 3 ijms-22-05096-f003:**
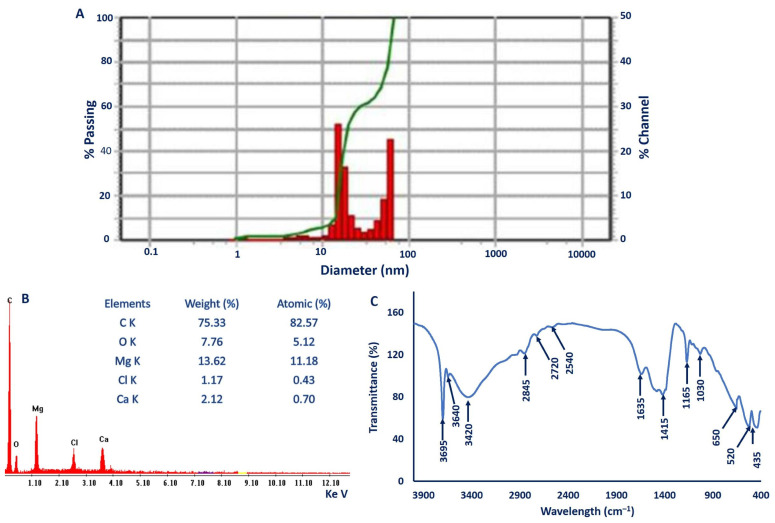
Characterization of biosynthesized MgO-NPs. (**A**) Dynamic light scattering (DLS) analysis, (**B**) energy dispersive X-ray (EDX) analysis, and (**C**) Fourier transform infrared (FT-IR) spectroscopy.

**Figure 4 ijms-22-05096-f004:**
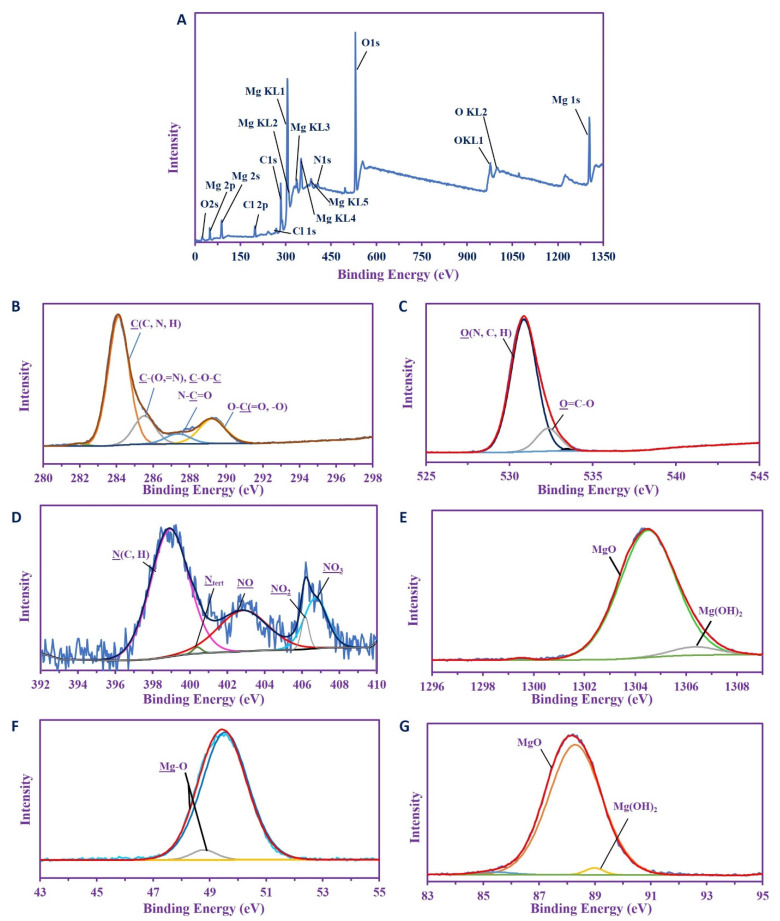
XPS analysis of biosynthesized MgO-NPs. (**A**) Overall survey, (**B**) C 1s, (**C**) O 1s, (**D**) N 1s, (**E**ȓ**G**) Mg 1s, Mg 2p, and Mg 2s, respectively.

**Figure 5 ijms-22-05096-f005:**
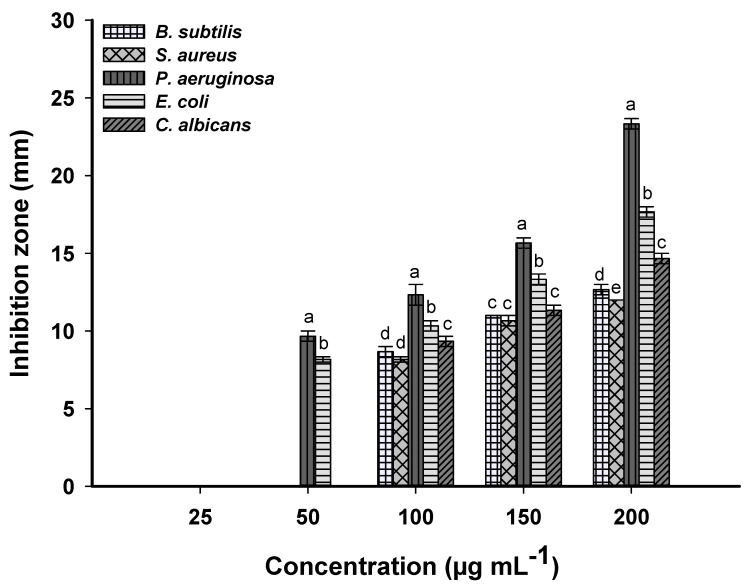
Antimicrobial activity of biosynthesized MgO-NPs against pathogenic Gram-positive and Gram-negative bacteria, and unicellular fungi. Different letters (a, b, c, d and e) on bars at the same concertation denote that mean values are significantly different (*p* ≤ 0.05) (*n* = 3).

**Figure 6 ijms-22-05096-f006:**
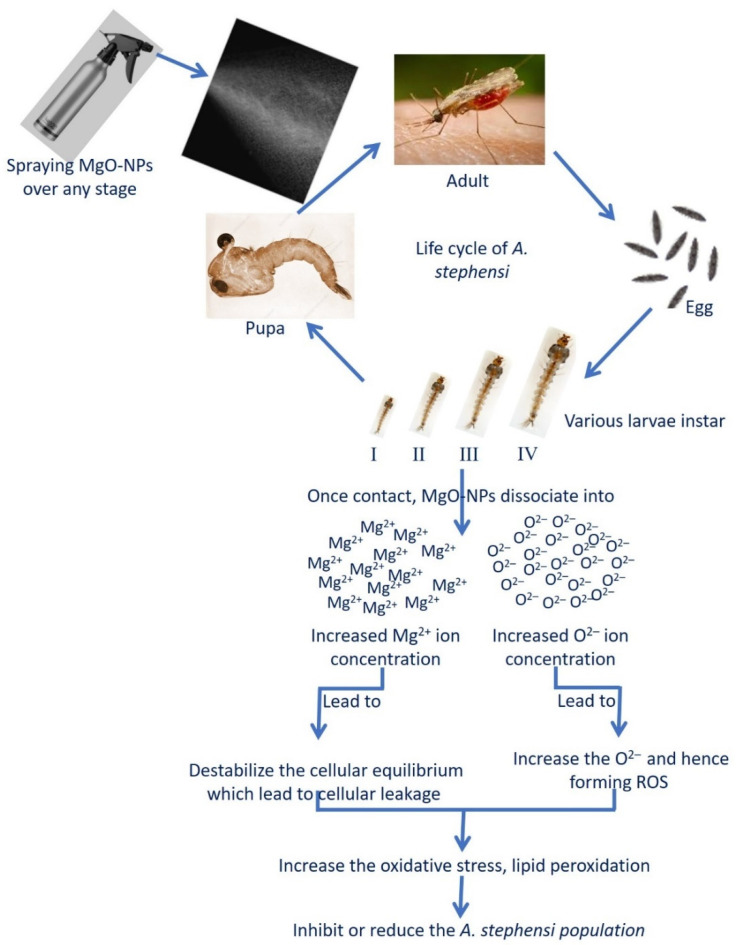
Prospective mechanisms of biosynthesized MgO-NPs as a mosquitocidal agent.

**Table 1 ijms-22-05096-t001:** The toxicity of biosynthesized MgO-NPs against *A. stephensi* larvae and pupa.

Target Instar	Mortality Percentages (%) ± SD *					LC50	LC90
	5 ppm	10 ppm	15 ppm	20 ppm	25 ppm		
I	30.2 ± 1.1	48.6 ± 1.7	67.6 ± 2.5	79.2 ± 2.2	91.8 ± 2.4	12.4	22.3
II	29.6 ± 1.1	41.8 ± 1.1	64.2 ± 1.9	75.6 ± 1.1	88.2 ± 1.6	13.10	23.5
III	27.8 ± 1.3	41.4 ± 1.8	59.8 ± 2.9	72.4 ± 2.9	81.4 ± 0.9	13.93	25.09
IV	24.4 ± 1.9	35.6 ± 2.3	51.4 ± 1.5	67.6 ± 2.6	72.8 ± 2.6	15.5	27.99
Pupa	22.8 ± 1.9	34.8 ± 1.3	48.2 ± 1.5	61.8 ± 3.5	69.2 ± 2.8	16.5	29.8

* Mortality was expressed as mean ± SD (standard deviation) of five replicates (*p* < 0.05) and calculated after 24 h. No mortality was observed in the control.

**Table 2 ijms-22-05096-t002:** The repellent assay of different concentrations of biosynthesized MgO-NPs against *A. stephensi.*

**Concentrations of MgO-NPs** (mg cm^−2^)	Repellent Percentages (%) ± SD							
	15 min	30 min	60 min	90 min	120 min	150 min	180 min	210 min
1	100.0 ± 0.0	100.0 ± 0.0	100.0 ± 0.0	100.0 ± 0.0	100.0 ± 0.0	69.3 ± 1.6	57.2 ± 1.1	43.1 ± 1.6
2.5	100.0 ± 0.0	100.0 ± 0.0	100.0 ± 0.0	100.0 ± 0.0	100.0 ± 0.0	72.4 ± 1.5	64.2 ± 1.6	49.4 ± 1.8
5	100.0 ± 0.0	100.0 ± 0.0	100.0 ± 0.0	100.0 ± 0.0	100.0 ± 0.0	100.0 ± 0.0	76.6 ± 1.2	67.6 ± 1.4
7.5	100.0 ± 0.0	100.0 ± 0.0	100.0 ± 0.0	100.0 ± 0.0	100.0 ± 0.0	76.6 ± 1.6	70.0 ± 1.6	50.6 ± 1.4
10	100.0 ± 0.0	100.0 ± 0.0	100.0 ± 0.0	100.0 ± 0.0	100.0 ± 0.0	80.9 ± 1.9	74.9 ± 1.3	59.6 ± 1.5

## Data Availability

The data presented in this study are available on request from the corresponding author.

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
