# Peer review of "An Eco-Friendly Approach to the Control of Pathogenic Microbes and Anopheles stephensi Malarial Vector Using Magnesium Oxide Nanoparticles (Mg-NPs) Fabricated by Penicillium chrysogenum"

_ijms, 2021, doi:10.3390/ijms22105096_

Round 1

Reviewer 1 Report

Fouda et al describe An eco-friendly approach to the control of pathogenic microbes and Anopheles stephensi malarial vector using magnesium oxide nanoparticles (Mg-NPs) fabricated by Penicillium chrysogenum. Need to be added references in most of the statement and methodology. Also need to proofread to eliminate the typo error.

After considering the following comment, this paper can be published.

-7–40nm, 12.5–15.5ppm, 150min  -spacing. Check all spacing in the units.

-Line 47-References

doi: 10.1007/s42770-019-00108-z

 doi: 10.1016/j.carbpol.2019.115228

Line 72-reference-doi: 10.2174/1872210514666200313121953.

Line 87-The as-formed MgO-rephrased

Line-95-The

Line-Reference-for the isolation

Line-119-Rae you sure this is the first report.

Line 141-metabolites or cell-free extract?. You have not isolated and purified the metabolites?

Line-the>>>>The-check all after fullstop.

Line-228 Phyto->>>phyto-

Line 226-228-Dose these atoms play a role in the synthesis of MgO-NPs?

 Line295-296-Repharsed to make better understanding.

Line 323-≈16.9 ? 7.5 nm >>>> ~16.9 ? 7.5 nm

Line 528- I am sure Muller Hinton agar media was not used for the C. albicans cultivation.

Antimicrobial activity. Reference for antimicrobial assays.

Line 408-prospective >>>>Prespective

Line-562/574: Check ? mark in the equations 4 and 5.

Figure-5_Put arrow above 25 and 50 µg/ml to indicate the complete killing

Figure-6: How long exposure (I mean spray time) by MgO-NPs

Make a high-resolution Figure-6.

Author Response

Thank you very much for reviewing my manuscript. Please see the attachment. 

Reviewer 2 Report

Green synthesis is a very promising and widely used technology to produce nanoparticles. I appreciate that the authors did a lot of measurements and analysis. Also, antimicrobial activity is widely described. I have a few comments on the article:
1.    Authors should unify the notation of units (the gap between number and unit) – page 1, line 32 and 37: 4.7±0.6mm; page 1, line 35: 5 mg/cm2
2.    Page 2, line 48 – “top-down. the” - capital letter after the dot.
3.    Page 2, line 55 – Please correct the sentence: “On the contrary, the biological route is characterized by it is a simple, rapid, eco-friendly, low-cost method, scalability, and biocompatibility as compared to chemical and physical processes [5].” May by the part: “On the contrary, the biological route is characterized by it is a simple, rapid….”.  But what about “…scalability, and biocompatibility…”?
4.    In the Introduction authors should cite the following papers:
•    Synthesis of Ag nanoparticle using R. officinalis, U. dioica and V. vitis-idaea extracts. doi: 1016/j.matlet.2019.04.027
•    Effect of P. kessleri extracts treatment on AgNPs synthesis. doi: 1080/24701556.2020.1726388
which will widen the information about the green synthesis of NPs by plant and algae extracts and
•    Preparing, Characterization and Anti-Biofilm Activity of Polymer Fibers Doped by Green Synthesized AgNPs. doi: 10.3390/polym13040605
which shows the possibilities of preparing a polymer composite doped with NPs and its use as an antimicrobial and antibiofilm agents
5.    Somewhere you put a dot at the end of the title: “2. Results and discussion.” and somewhere you don't: “2.1. Isolation and identification of the fungal isolate”.
6.    Page 3, line 95 – the sentence should begin with a capital letter.
7.    Page 3, line 101 – it should be agar media not “ager media”
8.    Page 4, line 29 – the sentence begins with a capital letter – Cultural, …
9.    Page 4, line 135-137 - Please consider changing the sentence as follows: Fungi are considered a promising tool for the green synthesis of metal and metal oxides NPs due to their reservoir of bioactive compounds. Fungi also tolerate and accumulate a high concentration of metals, and they are characterized by easy handling, biocompatibility, and scalability [4, 30].
10.    Page 5, line 154 - (Figure 2A). UV-vis is in Figure 2A
11.    On which day or hour of the synthesis was the UV-vis measured? Was synthesis fast (min, hours, or days)?
12.    Page 5, line 159-161 – Please correct the sentence, the verb is missing? 
13.    According to SPR what size and shape of NPs do authors suggest?
14.    Page 5, line 165 - (Figure 2B). XRD is in Figure 2B, not 3.
15.    Why authors mentioned the biological activities in this section (2.3.3. Transmission Electron Microscopy (TEM))? It doesn't belong here.
16.    Authors should show a better TEM micrograph of NPs.
17.    Sometimes Figure is in bold (Fig. 2C and D) and sometimes not (Figure 3A). Sometimes Fig. and sometimes Figure. Please unify it.
18.    Page 7, line 224 – Capital letter - The presence… The same -  page 7, line 248 and 249
19.    Based on FTIR authors should conclude which functional groups are responsible for the reduction and which are the capping agents.
20.    Page 7, line 224 – Capital letter Figure 6. Prospective. Please correct all text.
21.    Page 18, line 562 – There is “?” in equation (4).
22.    Authors should describe the mechanism of MgO-NPs toxicity. Is it the same as in the case of AgNPs? 

I recommend publishing the paper after a revision and accepting all comments.

Author Response

(The authors gave the same response as above.)
